# Instabilities of heavy magnons in an anisotropic magnet

Xiaojian Bai [1,2,3,8] ✉, Shang-Shun Zhang [4,8] ✉, Hao Zhang [4,5], Zhiling Dun [2], W. Adam Phelan [6], V. Ovidiu Garlea [1], Martin Mourigal [2] & Cristian D. Batista [4,7]

The search for new elementary particles is one of the most basic pursuits in physics, spanning from subatomic physics to quantum materials. Magnons are the ubiquitous elementary quasiparticle to describe the excitations of fully-ordered magnetic systems. But other possibilities exist, including fractional and multipolar excitations. Here, we demonstrate that strong quantum interactions exist between three flavors of elementary quasiparticles in the uniaxial spin-one magnet $FeI_2$. Using neutron scattering in an applied magnetic field, we observe spontaneous decay between conventional and heavy magnons and the recombination of these quasiparticles into a super-heavy bound-state. Akin to other contemporary problems in quantum materials, the microscopic origin for unusual physics in $FeI_2$ is the quasi-flat nature of excitation bands and the presence of Kitaev anisotropic magnetic exchange interactions.

The concept of quasiparticles is central to understand and predict the properties of condensed matter. For example, the quantization of collective atomic vibrations and spin precessions in long-range ordered solids[1] leads to the familiar concepts of phonons and magnons. When motion is harmonic, these bosonic excitations are free[2] and manifest in spectroscopic measurements as bands with well-defined energy-momentum dispersion. Interactions between phonons underpin many basic phenomena ranging from the anharmonic behavior of crystals and the lattice conductivity of thermoelectrics[3] to the rich excitation spectrum of liquid $^4He$[4]. In magnetism, interactions between magnons[5] can yield finite lifetimes by spontaneous (non-thermal) decay into multi-magnon states[6], resulting in incoherent excitation bands. Magnon decay is reminiscent of elementary particle decay, a ubiquitous quantum phenomenon of the Standard Model of subatomic physics. Although magnon instabilities are expected for a broad class of models[7–9], their experimental observation is rare and so far limited to a handful of quantum paramagnets[10,11] and non-collinear spin systems[12–16]. As

the search for quantum spin-liquids and their fractional excitations intensifies[17], achieving a quantitative understanding of magnon interactions is a pressing issue[18,19].

Unlike the Standard Model, where all elementary particles emerge from a single vacuum, a rich quasiparticle landscape[20] arises from distinct vacua (ground states) in the innumerable magnetic solids. In this context, it is surprising that decay instabilities have only been investigated in detail when magnon quasiparticles and their decay products carry the same fundamental quantum spin number: $\Delta S^z = \pm 1$ about the local quantization axis imposed by the underlying magnetic order. A counter-intuitive framework to realize strong magnon interactions is a system with large spin ($S \geq 1$) and strong single-ion and magnetic exchange anisotropies. We illustrate this concept in Fig. 1 using a ferromagnetic spin array. A $S = 1/2$ system (Fig. 1a) only admits single magnons (SMs) as elementary excitations, which carry a dipolar quantum number $|\Delta S^z| = 1$. Composite excitations of multiple (free or bound) SMs are possible, but they are not elementary; understanding their interactions often

[1]Neutron Scattering Division, Oak Ridge National Laboratory, Oak Ridge, TN 37831, USA. [2]School of Physics, Georgia Institute of Technology, Atlanta, GA 30332, USA. [3]Department of Physics and Astronomy, Louisiana State University, Baton Rouge, LA 70803, USA. [4]Department of Physics and Astronomy, University of Tennessee, Knoxville, TN 37996, USA. [5]Materials Science and Technology Division, Oak Ridge National Laboratory, Oak Ridge, TN 37831, USA. [6]PARADIM, Department of Chemistry, The Johns Hopkins University, Baltimore 21218 MD, USA. [7]Neutron Scattering Division and Shull-Wollan Center, Oak Ridge National Laboratory, Oak Ridge, TN 37831, USA. [8]These authors contributed equally: Xiaojian Bai, Shang-Shun Zhang. ✉e-mail: xbai@lsu.edu; shangshun89@gmail.com

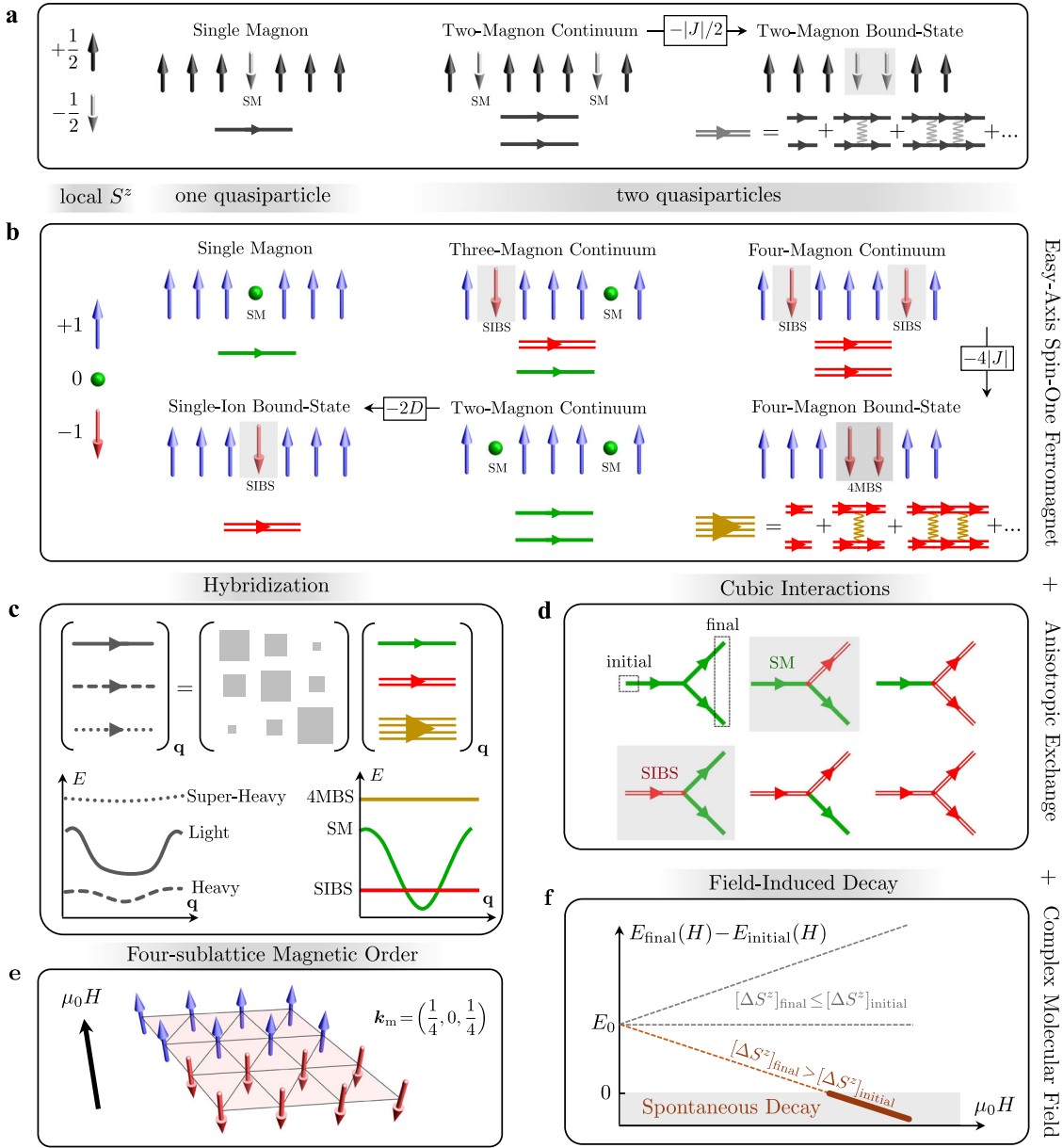

**Fig. 1 | Magnon hybridization, binding and field-induced decay in a uniaxial spin-1 system. a** The elementary excitations of a ferromagnetic array of $S = 1/2$ spins are single-magnon (SM) modes carrying a $\Delta S^z = -1$ quantum number. **b** For $S = 1$ spins with uniaxial anisotropy, elementary SM excitations coexist with a distinct quasiparticle called single-ion bound-state (SIBS) carrying $\Delta S^z = -2$. For $D \gg |J|$, the SIBS is an infinitely lived elementary excitation as the continuum of two free SMs is unstable. In that sense, the SIBS has a two-magnon character. The two quasiparticles sector comprises all possible combinations of free SM and SIBS elementary excitations and their non-perturbative bound states stabilized by short-range ferromagnetic interactions. This leads to a long-lived four-magnon bound-state (4MBS). **c** In FeI$_2$, at least three flavors of excitations overlap in momentum-energy space: dispersing SM and quasi-flat SIBS and 4MBS excitations. Spin non-conserving exchange interactions hybridize these excitations, giving rise to renormalized dispersion curves: SM form wide bands, SIBS narrow bands, and 4MBS are quasi-flat; hence the names of light, heavy and super-heavy

quasiparticles, respectively. **d** All possible cubic interaction vertices between initial one-quasiparticle and final two-quasiparticle states for a $S = 1$ system. The green, single lines (red, double lines) represent propagators for the SM and SIBS, respectively. The processes highlighted are the relevant magnon decay channels here. **e** Magnetic structure of FeI$_2$. **f** Effect of a magnetic field on the kinematic condition for decay. The Zeeman shift of a given state $\alpha$ depends on the total spin quantum number $[\Delta S^z]_\alpha$ as $E_\alpha(H) = -g\mu_B\mu_0 H[\Delta S^z]_\alpha$ where $g = 3.8(5)$. Decays are kinematically allowed if $E_{\text{final}}(H) - E_{\text{inital}}(H) \leq 0$. For interactions that conserve spin states, such as Heisenberg exchange, a magnetic field cannot change the net kinematic balance if decay conditions are not met in zero field. In contrast, spin non-conserving exchange interactions couple initial and final states with different quantum spin numbers. The positive energy offset in zero magnetic field, $E_{\text{final}}(0) - E_{\text{inital}}(0) \equiv E_0$ can be compensated by the differential Zeeman shift in finite fields, thereby activating spontaneous decay above a threshold field.

requires a non-perturbative treatment. To date, the vast majority of magnon decay studies have focused on the interactions between SMs and their multi-particle states. For a $S = 1$ system (Fig. 1b), the enlarged local Hilbert space yields a second type of on-site excitation where the same spin is flipped twice, from $S^z = +1$ to $-1$. This excitation is known as "single-ion bound-state" (SIBS) and becomes a

distinct elementary quasiparticle, with quantum number $|\Delta S^z| = 2$, for strongly uniaxial systems[21,22]. Naturally, composite excitations formed by multiple SMs and/or SIBSs are also possible, for instance four-magnon bound-states (4MBS) comprising two SIBS bound by short-range ferromagnetic exchange interactions (see Fig. 1b-right). SIBS and their composite excitations are fundamentally different

from SMs as they carry a multipolar quantum number $|\Delta S^z| = 2, 3, 4, \ldots$ and form quasi-flat bands that are in principle invisible to spectroscopic tools restricted by the dipole selection rule. An opportunity to observe these excitations, however, stems from their possible hybridization with conventional quasiparticles such as phonons or (single) magnons. The former mechanism is realized in $UO_2$[23], while the latter was recently uncovered and understood for $FeI_2$[24], which is the subject of this work.

$FeI_2$ is a quasi-2D Van der Waals material that comprises perfect *ab*-plane triangular layers of $Fe^{2+}$ ions surrounded by $I^-$ ligands (see "Methods" for a detailed description). At low temperature, the $Fe^{2+}$ ions bear effective $S_{eff} = 1$ magnetic moments with an easy-axis anisotropy $D \approx 2$ meV[25] along the crystallographic *c* axis. Magnetic exchange interactions are frustrated[26] with a ferromagnetic nearest-neighbor coupling $J_1 \approx -0.1D$ competing with weaker antiferromagnetic further-neighbor exchanges within and between the triangular planes[24] (see Supplementary Fig. 1). This competition stabilizes a striped antiferromagnetic (AF) order below $T_N = 9.5$ K[26,27], with rows of ferromagnetically aligned spins arranged in ↑↑↓↓ domains within the triangular layers, each with four magnetic sublattices (see Fig. 1e and Supplementary Fig. 2). At $T = 1.8$ K, the AF phase is stable up to a magnetic field of $\mu_0 H_1 = 4.8$ T before evolving into a complex sequence of ferrimagnetic phases below magnetic saturation at $\mu_0 H_{sat} = 12.5$ T[26]. Early neutron spectroscopy experiments[28] in the AF phase of $FeI_2$ elucidated that SIBS excitations, previously identified by infrared spectroscopy[29], form quasi-flat bands that lie below SM branches throughout the Brillouin zone. Recent quantitative studies[24,30] have demonstrated that off-diagonal components of the nearest-neighbor exchange interaction are responsible for the high degree of hybridization between overlapping dipolar (SM) and multipolar (SIBS, 4MBS, ...) excitations. These interactions can be parameterized using the *spin-non-conserving* terms $S_i^z S_j^+$ and $S_i^+ S_j^+$ of energy scale $J^{z\pm} \approx 1.1 J_1$ and $J^{\pm\pm} \approx 0.7 J_1$, respectively, or using an extended Kitaev-Heisenberg model[31]. In this context, all observed magnetic excitations in $FeI_2$ are hybrid between dipolar and multipolar quasiparticles; for simplicity, we will refer to them according to the character of their dominant quasiparticle.

## Results

The delicate balance of microscopic interactions and anisotropies in $FeI_2$ leads to a unique situation where distinct elementary quasiparticles, and their bound-states, overlap in momentum-energy space and strongly hybridize. This renormalizes their dispersion curves producing light (wide band) and heavy (narrow-band) quasiparticles (Fig. 1c). Given the regime of strong quantum interactions in $FeI_2$, it is natural to wonder if spontaneous decays are also possible. The most straightforward mechanism is through the *spin-non-conserving* exchange interactions because these activate cubic decay processes (Fig. 1d). For example, the $J^{\pm\pm} S_i^+ S_j^+$ term connects initial and final states whose quantum spin numbers differ by two. But two additional conditions are necessary to observe spontaneous decay. First, the six decay vertices of Fig. 1d must connect initial one-quasiparticle states to final two-quasiparticle states with a non-zero matrix element. As two (resp. three) combinations of quasiparticles exist for the initial (resp. final) states, this opens up many distinct decay channels. Second, decays must obey the conservation of total energy and crystal momentum. The fulfillment of these kinematic conditions depends on details of the excitation spectra amenable to external control, for instance, with a magnetic field. Kinematic decay conditions are not met in $FeI_2$ in the absence of a magnetic field as all multi-particle continua are just above the dominant excitation branches. However, the relative Zeeman shift between initial and final states with different quantum spin numbers can, as we will observe below, overcome this discrepancy and activate decay processes for an adequate magnetic field range (Fig. 1f).

To search for quasiparticle decay in $FeI_2$, we apply a magnetic field perpendicular to the triangular planes to tune the relative position of magnetic excitations within the AF phase (Fig. 1f) and examine the resulting momentum- and energy-resolved response using inelastic neutron scattering (see Methods). A slight misalignment between the magnetic field direction and the *c*-axis of our high-quality multi-gram crystal selects a single magnetic domain (see Supplementary Fig. 3, which dramatically simplifies interpretation of our results. In Fig. 2, we compare neutron-scattering data for $\mu_0 H = 0, 1, 2, 3$ and 4 T with SU(3)-generalized linear spin-wave (GLSW) calculations[32] for the exchange interactions of ref. 24 and $g = 3.8(5)$ (see Table 1). For $\mu_0 H \leq 2$ T, the number, dispersion, intensity, linewidth, and field-dependence of all the observed modes are in excellent agreement with GLSW predictions for all measured momenta (Fig. 2a, see also Supplementary Figs. 4–5 for more cuts). These spectra reflect the magnetic field evolution of eight modes: a SM and a SIBS for each of the four magnetic sublattices. Half of the modes have weak intensity for the momenta shown in Fig. 2. Excitations of the spin-down ferromagnetic stripes, corresponding to $\Delta S^z = +1$ and $+2$, experience a negative Zeeman shift $\Delta E_{Zeeman} = -g\mu_B \mu_0 H \Delta S^z$, and vice-verse for excitations of the spin-up stripes. The enhanced splitting of the $\Delta S^z = \pm 2$ magnon bound-states enables their unambiguous spectroscopic identification[33,34], see arrows on Fig. 2a, b.

While all excitation branches are sharp below $\mu_0 H \leq 3$ T, a striking deviation from GLSW predictions is observed at $\mu_0 H = 4$ T where the single-ion bound-state (SIBS) broadens considerably in the middle of the Brillouin zone, see yellow box in Fig. 2b. The line cut in Fig. 2c confirms the significant energy broadening of the SIBS peak at 3.76(1) meV with a full-width at half maximum (FWHM) of 0.38(1) meV, and reveals an anomalous energy width of 0.57(1) meV for the proximate single-magnon excitation at 4.41(3) meV, see Supplementary Table 1 for fit results. In line cuts for other momenta and fields, all branches appear resolution-limited with a FWHM of $\approx 0.20$ meV. We tentatively ascribe these characteristic features to the activation of decay processes for both the SM and SIBS quasiparticles.

Further evidence for strong magnon interactions in $FeI_2$ comes from the observation of four-magnon (4MBS) and six-magnon bound-states (6MBS) in magneto-optics[30]. These higher-order exchange bound-states are stabilized by the narrow-band of the system and the presence of ferromagnetic interactions at short distances in a given stripe of the underlying magnetic structure (Fig. 1e). 4MBS excitations are clearly visible in our neutron-scattering data at $\mu_0 H = 3$ T as weak and non-dispersing modes, unaccounted for by GLSW. Of particular interest is the 2.5 meV mode indicated by a white arrow in Fig. 2b: it lies below the SM branch observed at 3.0 meV in Fig. 2c but predicted by GLSW at 2.8 meV, indicative of mode repulsion (see Supplementary Fig. 4 for more examples of this behavior). At $\mu_0 H = 4$ T, the 4MBS excitation moves down in energy but the shift of the SM peak persists. In spite of their hexadecapolar nature ($\Delta S^z = 4$), 4MBS excitations are detected in our experiment because of their strong hybridization to dipolar fluctuations[30]: given their impact on the SM branch they must be treated on equal footing as a distinct quasiparticle flavor.

To explain the anomalous mode broadening uncovered by our experiments in finite magnetic field, we refine our previous SU(3)-generalized spin-wave theory using a perturbative expansion that accounts for quasiparticle interactions at the one-loop level. To capture the hybridization, energy renormalization, and decay rate of the SM and SIBS excitations, it is sufficient to retain cubic interaction vertices that couple the one- and two-quasiparticle sectors, i.e., we drop the negligible contribution from quartic vertices, see Methods for full details. The essential results of these non-linear calculations (GNLSW) are shown for $\mu_0 H = 3$ T and 4 T in Fig. 3a, b. The gray and colored regions indicate the continua of allowed energies and

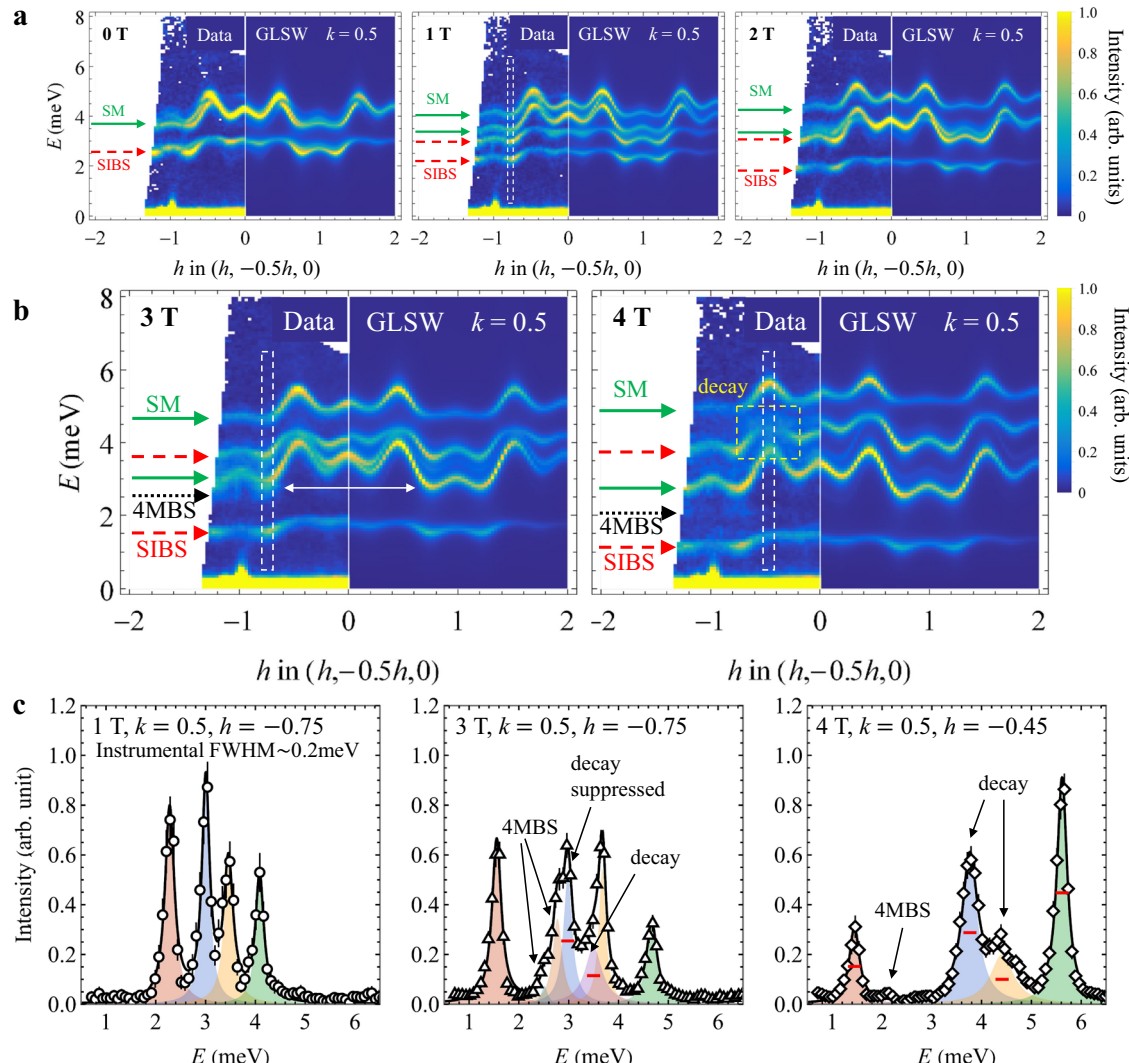

**Fig. 2 | Field-induced magnon instabilities in neutron-scattering spectra of FeI₂.** **a** Momentum- and energy-resolved neutron-scattering spectra of FeI₂ at $T = 1.8$ K (AF phase, single-domain) and $\mu_0H = 0$, 1, and 2 T (below all decay threshold) with excellent match to generalized linear spin-wave (GLSW) predictions accounting for excitations with both single-magnon (SM, solid green arrows) and single-ion bound-states (SIBS, dashed red arrows) dominant character. The intensity scale is in arbitrary units. The momentum direction corresponds to $\mathbf{Q} = (h, 1/2 - h/2, 0)$ with perpendicular directions integrated over $|\Delta k, \ell| \leq 0.05$ r.l.u. **b** Momentum-energy slices for $\mu_0H = 3$ and 4 T (above decay threshold) revealing deviations from GLSW predictions. The white double-sided arrow signals the presence of an additional

excitation at 3 T, consistent with a 4-magnon bound-state (4MBS, black arrow). The yellow dashed box highlights a considerable energy broadening for otherwise sharp excitations at 4 T, the hallmark of spontaneous magnon decay. **c** Energy lineshape for selected excitations from constant-$\mathbf{Q}$ cuts through the above data (open symbols) integrated over $|\Delta h| \leq 0.1$ r.l.u., see the white dashed regions in panels **a** and **b**. Errorbars correspond to one standard deviation. Lorentzian peak fits (black curves for overall fits and colored shaded areas for individual peaks) at various magnetic fields highlighting departure from the resolution limit (red bars, FWHM ≈ 0.2 meV, obtained from fits at $\mu_0H = 1$ T), i.e., field-induced magnon decay. Fit results are reported in Supplementary Table 1 and Supplementary Fig. 6.

momenta for each possible combination of two unbound SM or SIBS quasiparticles. Decays are kinematically allowed where a given excitation branch overlaps with one or several of these shaded regions, with the larger decay rates (red shading in Fig. 3a, b central panels) originating from the colored continua.

For $\mu_0H = 4$ T, Fig. 3b, our GNLSW calculations predict large decay rates for the top of the $E_4$ and $E_6$ bands (see band labeling in

Fig. 3). This yields a broadened neutron-scattering response highlighted by the dashed yellow box in Fig. 3b, in excellent agreement with our experimental observations. While the hybrid character of all the bands is fully accounted for in our quantitative decay rate calculations, it is instructive to focus on their dominant character at a given wave-vector to elucidate their decay mechanism. The broadening of band $E_4$ around 3.8 meV stems from the emission of a $\Delta S^z = +2$ SIBS on branch $E_2$ by a $\Delta S^z = +1$ SM that correspondingly looses energy and momentum. The broadening observed around 4.4 meV for band $E_6$, corresponds to a $\Delta S^z = -2$ SIBS decaying into two SM excitations: one at the bottom of the $E_2$ band with $\Delta S^z = +1$ and one at a different wave-vector of the $E_6$ band where the $\Delta S^z = -1$ character dominates. These decay processes correspond to the spontaneous creation and annihilation of a single-ion bound-state through a net change of two units of angular momentum, implying that the relevant interaction vertices are mediated by the

**Table 1 | Hamiltonian parameters of FeI₂ (meV)**

| Nearest-neighbor | | | Further neighbor | | | | | | Single-ion |
|---|---|---|---|---|---|---|---|---|---|
| $J_1^{\pm}$ | $J_1^{\pm\pm}$ | $J_1^{z\pm}$ | $J_2^{\pm}$ | $J_3^{\pm}$ | $J_0'^{\pm}$ | $J_1'^{\pm}$ | $J_{2a}'^{\pm}$ | – | |
| −0.236 | −0.161 | −0.261 | 0.026 | 0.166 | 0.037 | 0.013 | 0.068 | – | |
| $J_1^{zz}$ | – | – | $J_2^{zz}$ | $J_3^{zz}$ | $J_0'^{zz}$ | $J_1'^{zz}$ | $J_{2a}'^{zz}$ | | $D$ |
| −0.236 | – | – | 0.113 | 0.211 | −0.036 | 0.051 | 0.073 | | 2.165 |

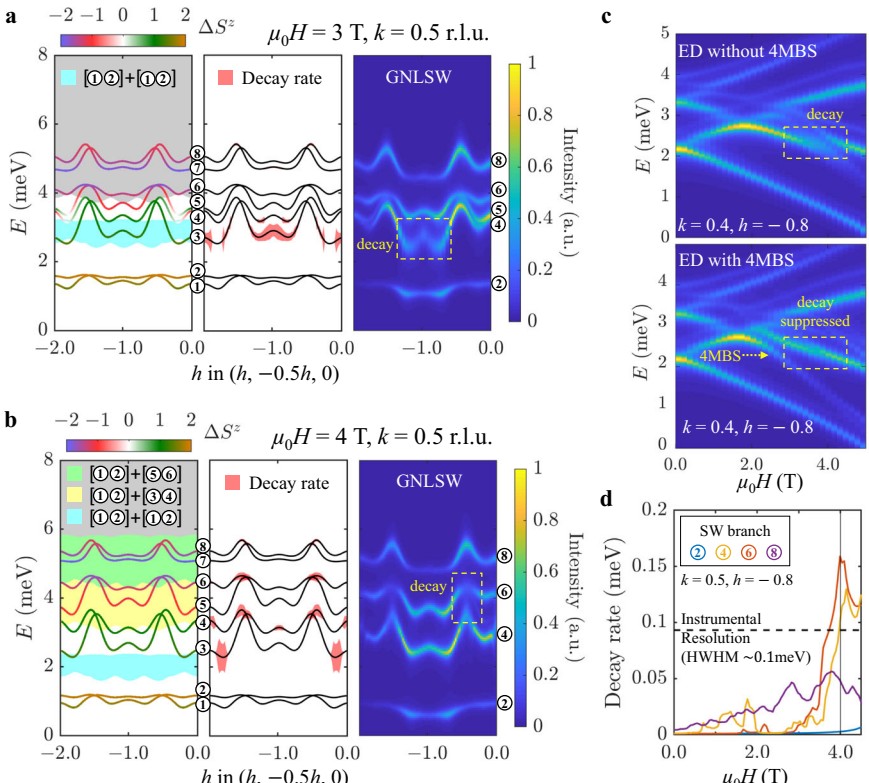

**Fig. 3 | Quantum magnon dynamics captured by one-loop expansion and exact diagonalization. a, b** Predictions from SU(3)-generalized non-linear spin-wave theory with one-loop order corrections (GNLSW) for the Hamiltonian of FeI$_2$ along the experimental momentum-energy slices of Fig. 2 at 3 T and 4 T, respectively. Each panel shows in turn (left to right): kinematic conditions for decay, predicted decay rate, and realistic neutron-scattering intensity. For a single-domain of the AF structure, eight hybridized bands ($E_n$) are present: one SM and one SIBS for each of the four magnetic sublattices. These bands are numbered and color-coded to reflect their $\Delta S^z$ value, which changes as a function of momentum transfer. Shaded regions indicate the extent of the two-quasiparticle continua that can be constructed from these eight quasiparticles, with a color shade (resp. gray shade) for states which do (resp. do not) yield significant decay rates. For instance, the cyan region corresponds to a 2-selection among branches $E_1$ and/or $E_2$. For the chosen cut direction, branches with large decay rates (red shading) may not have large spectral weight to be apparent in the calculated neutron-scattering intensity. **c** Magnetic field evolution of excitations calculated from Exact Diagonalization (ED) for the Hamiltonian of FeI$_2$ on a finite cluster of $5 \times 5 \times 5$ unit cells (500 spins) with a Hilbert space truncated to include (bottom) or not include (top) up to 4-magnon excitations. The finite-size calculation restricts the set of accessible momenta such that ED plots are for a momentum proximate to that of Fig. 2c, and slightly modifies the kinematic decay conditions compared to GNLSW calculations. **d** Magnetic field-dependence of the GNLSW decay rate (Lorentzian half-width at half maximum) for various branches at selected momenta.

anisotropic, *spin-non-conserving* term $J^{\pm\pm} S_i^+ S_j^+$, see vertices with a gray background in Fig. 1d. Although this mechanism produces a finite lifetime for all excitation branches in the AF phase of FeI$_2$, the broadening only becomes visible in experiments when the decay rate exceeds the FWHM instrumental energy-resolution of around 0.2 meV, see Fig. 3d, Supplementary Fig. 6, and Supplementary Table 1. Thus spontaneous decays are only visible in our experiment in a narrow field range around $\mu_0 H = 4$ T, due to the narrow bandwidth of the lowest-energy branch essential to the decay processes.

Surprisingly, a qualitatively different phenomenon occurs for $\mu_0 H = 3$ T, Fig. 3a. While our GNSLW calculations predict strong decay for excitations at the bottom of the $E_4$ band, no visible broadening is observed in the experimental results of Fig. 2b, c. Instead, the putative unstable branch lies proximate to the 4MBS excitation discussed previously. Including this composite quasiparticle in our GNLSW calculations is impractical as it requires to sum ladder diagrams to infinite order in a perturbative loop expansion. We avoid this problem by performing an exact diagonalization (ED) of the SU(3) spin-wave Hamiltonian at quartic order on a finite lattice. Truncating the Hilbert space to only include up to two (free or bound) composite quasiparticles allows to reach adequate system sizes (see Methods). The quartic term is essential to

form a 4MBS from the continuum of two free SIBSs. Results without and with quartic vertices, Fig. 3c, explain the strong suppression of decays observed in our experiments as stemming from the finite probability of decay products to form a 4MBS instead of propagating independently in the system. At the microscopic level, this non-perturbative phenomenon, which we observe and understand for the first time, comes from the unique interplay between heavy SIBS quasiparticles, attractive (ferromagnetic) interaction at short distances, and spin-non-conserving terms.

## Discussion

Our neutron-scattering experiments on FeI$_2$ reveal a rich and field-tunable quantum many-body physics phenomenology that is quantitatively explained by our theory. We observe three distinct flavors of quasiparticles: light dipolar SM fluctuations, heavy quadrupolar SIBS quasiparticles, and super-heavy hexadecapolar 4MBS excitations (Fig. 1b) stabilized by attractive short-range interactions. These quasiparticles mix, decay and pair onto each other in a way reminiscent of high-energy particle physics. Our observations of spontaneous emission of a heavy quasiparticle by a magnon, the decay of the former into two free magnons, and the suppression of decay channels by the non-perturbative recombination of decay products

into super-heavy bound-states, are observed for the first time in the realm of condensed-matter systems. Within magnetism, our work challenges the conventional view that compounds with large spin and large uniaxial anisotropy behave classically. In fact, in $FeI_2$, this combination produces unique quantum magnon dynamics brought to light by spin-non-conserving off-diagonal exchange interactions. As such interactions are also essential to stablize quantum spin-liquids and their fractionalized excitations in Kitaev magnets[18], our work considerably broadens the range of quasiparticle phenomena expected in these spin-orbit coupled quantum magnets. The theoretical tools we have developed to understand $FeI_2$ apply to many other materials[16] and may be used in the future to sharpen our general understanding of large-spin magnets.

## Methods
### Crystal growth
Starting materials of Iron (≥99.98% purity from Alfa Aesar) and Iodine (≥99.99% purity from Alfa Aesar) were sealed in evacuated quartz tubes. The first synthesis step consists in a chemical vapor transport growth using a tube furnace (School of Physics, Georgia Tech) with the hot end at 570 °C and the cold end at room temperature[35] forming a collection of mm-size $FeI_2$ crystals. After grinding into fine powders in a glove box with water content ≤2 ppm, the crystal structure was checked using a PANAnalytical Empyrean Cu-$K\alpha$ diffractometer (Materials Characterization Facility, Georgia Tech) with samples loaded in an air-tight domed holder in the glove box. This confirmed the expected crystal structure and was consistent with results reported in ref. 24. Around 10 grams of the polycrystalline sample was sealed in a quartz tube under vacuum. The ampule was then placed in a graphite crucible attached to a rotator in a Ultrahigh Temperature Midscale Induction Bridgman/CZ Furnace (PARADIM facility, Johns Hopkins University). The crucible was passed through a hot zone of ≈ 600 °C with rotating speed 20 RPM/min and lowering rate 10 mm/hr. The growth yielded a large boule from which a 4.53 g high-quality crystal was extracted with clear c-axis facet. The resulting crystal was mounted on an aluminum sample holder sealed in a Helium-filled glove box. The holder was designed to keep the sample from moisture and oxygen contamination and had a small enough diameter to fit in the ⌀32 mm diameter of a cryomagnet. The mosaic of the crystal was checked with neutrons to be around ≤3°.

### Thermo-magnetic properties of $FeI_2$
$FeI_2$ crystallizes in the space-group $P\bar{3}m1$ with lattice parameters $a = 4.05$ Å and $c = 6.75$ Å at $T = 300$ K[36]. $FeI_2$ comprises triangular layers of $Fe^{2+}$ ions with magnetic interactions mediated by direct exchanges and super-exchanges through the $I^-$ ligands above and below the triangular plane. The combination of crystal-field and spin-orbit coupling effects on the $Fe^{2+}$ ions leads to effective $S = 1$ magnetic moments with an easy-axis anisotropy along the c-axis and several transitions to higher-energy multiplets above 25 meV[37]. In zero magnetic field, $FeI_2$ displays a long-range magnetic order below $T_N = 9.5$ K[26,27] through a first order transition with no apparent lattice distortion[27,38]. The magnetic structure is described by a propagation vector $\mathbf{k}_{AF} = (0, 1/4, 1/4)$ and the phase referred to as the "AF" phase given the absence of net magnetization. Within the triangular plane, the system forms a up-up-down-down stripe order shown in Supplementary Fig. 2.

Three types of magnetic domains are typically stabilized in zero magnetic field, related by 120° rotations[24] with propagation vectors $\mathbf{k}_{AF}^{(1)} = (0, 1/4, 1/4)$, $\mathbf{k}_{AF}^{(2)} = (-1/4, 0, 1/4)$ and $\mathbf{k}_{AF}^{(3)} = (1/4, -1/4, 1/4)$. When a magnetic field is applied along the crystallographic c-axis, a series of meta-magnetic transitions were observed in bulk magnetization measurements[39,40]. Associated magnetic structures were investigated using neutron diffraction[26] below the saturation magnetic field of

$\mu_0 H_s \approx 12.5$ T[39]. Below $T \approx 2$ K, the first magnetic transition occurs above $\mu_0 H_1 \geq 4.5$ T. The results presented here are restricted to magnetic fields $\mu_0 H \leq 4$ T and temperatures $T \leq 2$ K, such that the underlying magnetic structure for $FeI_2$ is AF. As explained in the main text and shown in Supplementary Fig. 3, this was checked by taking elastic cuts through the neutron-scattering data, which also revealed that a pre-dominantly single-domain magnetic state, corresponding to $\mathbf{k}_{AF}^{(1)}$, was stabilized in the sample.

### Neutron-scattering measurements
Inelastic neutron-scattering experiments were performed on the HYSPEC spectrometer at the Spallation Neutron Source (SNS), Oak Ridge National Laboratory (ORNL), USA[41]. The sample was mounted on a stick inserted in a $\mu_0 H_{max} = 8$ T vertical-field self-shielded superconducting magnet reaching a base temperature around $T = 1.8$ K. The sample was rotated around its c-axis over a range of 360° degrees in steps of 1° degree allowing a complete mapping of excitations in the scattering plane. The narrow out-of-plane coverage of ± 7° degrees of the magnet restricts the momentum transfer in the out-of-plane direction. All measurements were performed in unpolarized mode with an incoming neutron energy of $E_i = 9$ meV and Fermi choppers speed of 420 Hz yielding an elastic full-width at half-maximum energy-resolution on the sample of 0.20 meV. The center detector bank was positioned at a $2\theta$ angle of − 35°. Five magnetic field configurations were used corresponding to $\mu_0 H = 0, 1, 2, 3,$ and 4 T. When theoretical calculations are compared to experiments, the computed dynamical structure factors take into account all relevant experimental effects including magnetic form factor and neutron dipole factor.

### Data analysis
Data was reduced and analyzed in MANTID[42] on the SNS analysis cluster at ORNL. Symmetry operations that preserve the single-domain magnetic structure were applied to the data to increase statistics. Throughout the manuscript, the scattering intensity is measured as a function of energy transfer $E$ and momentum transfer $\mathbf{Q} = h\mathbf{a}^* + k\mathbf{b}^* + l\mathbf{c}^* \equiv (h, k, l)$ where $\mathbf{a}^*$, $\mathbf{b}^*$ and $\mathbf{c}^*$ are the primitive vectors of the triangular-lattice reciprocal space and $(h, k, l)$ are Miller indices in reciprocal lattice units. The usual convention that $\mathbf{a}^*$ and $\mathbf{b}^*$ make an 60° angle is used, see Supplementary Fig. 1.

### Model Hamiltonian
All theoretical calculations were performed using the zero-field exchange Hamiltonian obtained in ref. 24 including an uniaxial single-ion anisotropy $-D\sum_i (S_i^z)^2$ and exchange interactions up to third-neighbors in plane ($J_1$ to $J_3$) and out-of-plane ($J_0'$ to $J_2'$) as defined on Supplementary Figure 1. For nearest-neighbor bonds, all symmetry allowed diagonal and off-diagonal exchange interactions are taken into account, which yields the Hamiltonian

$$\mathcal{H}_{\text{n.n.}} = \sum_{\langle i,j \rangle} \left\{ J_1^{zz} S_i^z S_j^z + \frac{1}{2} J_1^{\pm} \left( S_i^+ S_j^- + S_i^- S_j^+ \right) + \frac{1}{2} J_1^{\pm\pm} \left( \gamma_{ij} S_i^+ S_j^+ + \gamma_{ij}^* S_i^- S_j^- \right) \right. $$
$$\left. - \frac{i J_1^{z\pm}}{2} \left[ \left( \gamma_{ij}^* S_i^+ - \gamma_{ij} S_i^- \right) S_j^z + S_i^z \left( \gamma_{ij}^* S_j^+ - \gamma_{ij} S_j^- \right) \right] \right\}, $$

where $\gamma_{ij} = e^{i\theta_{ij}}$ are bond-dependent phase factors with $\theta_{ij} = \theta_{ji} = 0, +\frac{2}{3}, -\frac{2}{3}$ depending on the direction of the bond of the triangular lattice[31].

For further-neighbor bonds, only diagonal anisotropy is considered, which yields the Hamiltonian

$$\mathcal{H}_{\text{f.n.}} = \sum_{(i,j)} \left\{ J_{\text{f.n.}}^{zz} S_i^z S_j^z + \frac{1}{2} J_{\text{f.n.}}^{\pm} \left( S_i^+ S_j^- + S_i^- S_j^+ \right) \right\} $$

for bonds $J_2 J_3 J_0' J_1'$ and $J_2'$.

In this work, we adopt the representative values of exchanges interactions for FeI$_2$ obtained in ref. 24 by joint fits to the zero magnetic-field energy-integrated data in the paramagnetic phase and the energy-resolved data in the magnetically ordered phase.

Although we will not use this notation in the present manuscript, we note that, alternatively, the nearest-neighbor exchange matrix for FeI$_2$ can be recast as an extended Kitaev-Heisenberg ($K$−$J$) model[31]

$$\mathcal{H}_{\text{n.n.}} = \sum_{\langle ij \rangle_\gamma} \Big[ J_1 \mathbf{S}_i \cdot \mathbf{S}_j + K_1 S_i^\gamma S_j^\gamma + \Gamma_1 \left( S_i^\alpha S_j^\beta + S_i^\beta S_j^\alpha \right)$$
$$+ \Gamma_1' \left( S_i^\gamma S_j^\alpha + S_i^\gamma S_j^\beta + S_i^\alpha S_j^\gamma + S_i^\beta S_j^\gamma \right) \Big],$$

with $J_1 = -0.41$ meV, $K_1 = 0.53$ meV, $\Gamma_1 = -0.02$ meV, and $\Gamma_1' = 0.01$ meV.

Finally, a Zeeman term $\mathcal{H}_{\text{Zeeman}} = -g\mu_B \mu_0 HS$ is included to account for the effect of magnetic field. The $g$-factor is obtained from GLSW fitting to the neutron-scattering data at low fields ($\mu_0 H \leq 2$T) with all exchange and single-ion parameters fixed, yielding a value of 3.8(5).

## Loop expansion

In zero field, FeI$_2$ has been successfully modeled by a SU(3) GLSW theory[24]. To explain the field-induced effects studied in this work, one must go beyond the GLSW and consider the interaction between the quasiparticles. Explicitly, we perform a systematic perturbation theory that corresponds to an expansion in the parameter $1/M$, where $M$ is the total number of SU($N$) bosons per site. Note that this expansion coincides with the well-known $1/S$ expansion for the particular case $N = 2$ ($M = 2S$ for $N = 2$). As will be shown below, the order of a given Feynman diagram of the expansion is determined by the number of independent loops, i.e., of closed lines of SU($N$) boson propagators.

To count the order of a given Feynman diagram, it is convenient to rescale the SU($N$) boson operator by a factor $1/\sqrt{M}$, namely $\beta_{i,m} = \beta'_{i,m}/\sqrt{M}$ (see the next section for explicit definition), $m = 1, 2, ..., N$. Consequently, $M$ becomes an overall prefactor of the rescaled Hamiltonian, $H = H'/M$. Since the original interaction vertices $V^{(n)} (n \geq 3)$, the coefficients of a triple product of the boson operators, scale as $M^{2-n/2}$, all vertices of the rescaled Hamiltonian $H'$ becomes of order $M$, while the propagator of a boson is still of order $1/M$. Therefore, the order $p$ of a particular one-particle irreducible diagram constructed by $V$ vertices and $I$ internal lines is $V - I$ (note that the frequency $\omega$ is of order $M^0$). Since the number of loops is $L = I - V + 1$, we obtain that the power $p = 1 - L$ is only determined by the number of loops in a particular Feynman diagram.

## Cubic vertex and self-energy

FeI$_2$ is described by an effective $S = 1$ spin model,

$$\mathcal{H} = \sum_{\langle ij \rangle} \sum_{\mu\nu} \hat{S}_i^\mu \mathcal{J}_{ij}^{\mu\nu} \hat{S}_j^\nu - D \sum_i \mathcal{Q}_i^{zz} - \sum_i h^\mu S_i^\mu, \tag{1}$$

where $\hat{S}_i^\mu, \mu = x, y, z$ is the spin-1 operator and the single-ion anisotropy term is proportional to the ($zz$) component of quadrupolar moment $\mathcal{Q}_i^{\mu\nu} = (\hat{S}_i^\mu \hat{S}_i^\nu + \hat{S}_i^\nu \hat{S}_i^\mu)/2 - 2/3$ (symmetric traceless components of $\hat{\mathbf{S}}_i \otimes \hat{\mathbf{S}}_i$). The spin-exchange tensor $\mathcal{J}_{ij}^{\mu\nu}$ is described in the main text.

To describe the magnetically ordered phase, it is convenient to work in the local reference frame defined by the $SU(3)$ rotation

$$\begin{pmatrix} \beta_{i,+1} \\ \beta_{i,0} \\ \beta_{i,-1} \end{pmatrix} = U_i^\dagger \begin{pmatrix} b_{i\uparrow} \\ b_{i0} \\ b_{i\downarrow} \end{pmatrix}, \tag{2}$$

where $U_i \in SU(3)$. The magnetic order corresponds to a macroscopic occupation of the $\beta_{i,+1}$ boson $\langle \beta_{i,+1} \rangle = \langle \beta_{i,+1}^\dagger \rangle \simeq \sqrt{M}$ and $M = 1$ for the case under consideration. Because of the strong single-ion anisotropy, we can safely assume that $\sum_{m\neq 1} \langle \beta_{i,m}^\dagger \beta_{i,m} \rangle \ll M$. This assumption justifies the $1/M$ expansion that was discussed in the previous section:

$$\beta_{i,+1}, \beta_{i,+1}^\dagger = \sqrt{M - \beta_{i,0}^\dagger \beta_{i,0} - \beta_{i,-1}^\dagger \beta_{i,-1}}$$
$$\simeq \sqrt{M} \left[ 1 - \frac{1}{2M} \sum_{m\neq 1} \beta_{i,m}^\dagger \beta_{i,m} - \frac{1}{8M^2} \sum_{m\neq 1} \left( \beta_{i,m}^\dagger \beta_{i,m} \right)^2 + \mathcal{O}\left(\frac{1}{M^3}\right) \right]. \tag{3}$$

The corresponding semi-classical expansion of the dipolar and quadrupolar operators are

$$\hat{S}_i^\mu = M \mathcal{S}_c^\mu(i) + \sqrt{M} \sum_{m\neq 1} \left( \mathcal{S}_{1m}^\mu(i)\beta_{i,m} + h.c. \right) + \sum_{m,n\neq 1} \mathcal{S}_{mn}^\mu(i)\beta_{i,m}^\dagger \beta_{i,n}$$
$$- \frac{1}{2\sqrt{M}} \sum_{m,n\neq 1} \left( \mathcal{S}_{1m}^\mu(i)\beta_{i,n}^\dagger \beta_{i,n}\beta_{i,m} + h.c. \right) + \mathcal{O}\left(\frac{1}{M^{3/2}}\right), \tag{4}$$

$$\mathcal{Q}_i^{zz} = M \mathcal{Q}_c^{zz}(i) + \sqrt{M} \sum_{m\neq 1} \left( \mathcal{Q}_{1m}^{zz}(i)\beta_{i,m} + h.c. \right) + \sum_{m,n\neq 1} \mathcal{Q}_{mn}^{zz}(i)\beta_{i,m}^\dagger \beta_{i,n}$$
$$- \frac{1}{2\sqrt{M}} \sum_{m,n\neq 1} \left( \mathcal{Q}_{1m}^{zz}(i)\beta_{i,n}^\dagger \beta_{i,n}\beta_{i,m} + h.c. \right) + \mathcal{O}\left(\frac{1}{M^{3/2}}\right), \tag{5}$$

where

$$\mathcal{S}_c^\mu(i) = \tilde{L}_{11}^\mu(i), \quad \mathcal{S}_{1m}^\mu(i) = \tilde{L}_{1m}^\mu(i), \quad \mathcal{S}_{mn}^\mu(i) = \tilde{L}_{mn}^\mu(i) - \tilde{L}_{11}^\mu(i)\delta_{mn},$$
$$\mathcal{Q}_c^{zz}(i) = \tilde{O}_{11}^{zz}(i), \quad \mathcal{Q}_{1m}^{zz}(i) = \tilde{O}_{1m}^{zz}(i), \quad \mathcal{Q}_{mn}^{zz}(i) = \tilde{O}_{mn}^{zz}(i) - \tilde{O}_{11}^{zz}(i)\delta_{mn}, \tag{6}$$

and $\tilde{L}^\mu(i) = U_i^\dagger L^\mu U_i, \tilde{O}^{zz}(i) = U_i^\dagger (L^z)^2 U_i$ with the matrices $L^\mu$ are the generators of the SO(3) group. The variables defined in Eq. (6) depend only on the sublattice index because of the translational symmetry of the magnetic structure.

By using the expansions (4) and (5), we obtain a generalized semi-classical expansion of the spin Hamiltonian

$$\mathcal{H} = \mathcal{E}^{(0)} + \mathcal{H}^{(2)} + \mathcal{H}^{(3)} + \mathcal{O}(M^0), \tag{7}$$

where $\mathcal{E}^{(0)}$ and $\mathcal{H}^{(2)}$ have been computed explicitly before[24]. We note that $\mathcal{E}^{(0)} \propto M^2$ and $\mathcal{H}^{(2)} \propto M$ according to the series expansion of the spin and quadrupole operators in Eqs. (4) and (5). Here, we will focus on the cubic term, $\mathcal{H}^{(3)}$, which is of $\mathcal{O}(\sqrt{M})$. The one-loop contributions from the quartic term, $\mathcal{H}^{(4)}$, correspond to a simple renormalization of the single-mode dispersion relation (real part of the self-energy), that is obtained by expressing $\mathcal{H}^{(4)}$ in normal ordering. The corresponding one-loop Feynman diagrams involving quartic vertexes that contribute to the single-particle self-energy are:

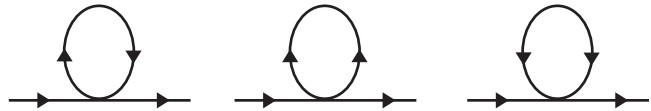

where the lines represent the propagators of the original bosons (before performing the Bogoliubov transformation). We note here that we have neglected these quartic contributions to the self-energy because they turn out to be very small ($10^{-3}J$, where $J$ represents the energy scale of the dominant exchange interaction) due the the large single-ion anisotropy.

The cubic interaction term is:

$$
\begin{aligned}
\mathcal{H}^{(3)} = \sum_{\langle ij \rangle} \sqrt{M} &\Bigg[ \sum_{m,n \neq 1} \left( V_1^m(i,j) \beta_{j,n}^\dagger \beta_{j,n} \beta_{j,m} + V_1^m(j,i) \beta_{i,n}^\dagger \beta_{i,n} \beta_{i,m} + h.c. \right) \\
&+ \sum_{l,m,n \neq 1} \left( V_2^{lmn}(i,j) \beta_{j,m}^\dagger \beta_{j,n} \beta_{i,l} + V_2^{lmn}(j,i) \beta_{i,m}^\dagger \beta_{i,n} \beta_{j,l} + h.c. \right) \Bigg] \\
&+ \frac{1}{2\sqrt{M}} \sum_i \sum_{m,n \neq 1} \left( (D\mathcal{Q}_{1m}^{zz}(i) + h^\mu \mathcal{S}_{1m}^\mu(i)) \beta_{i,n}^\dagger \beta_{i,n} \beta_{i,m} + h.c. \right),
\end{aligned}
$$

(8)

where $V_1^m(i,j) = -\frac{1}{2}\mathcal{S}_c^\mu(i)\mathcal{J}_{ij}^{\mu\nu}\mathcal{S}_{1m}^\nu(j)$, $V_2^{lmn}(i,j) = \mathcal{S}_{1l}^\mu(i)\mathcal{J}_{ij}^{\mu\nu}\mathcal{S}_{mn}^\nu(j)$. Translational invariance implies that $V_1^m(i,j) \equiv V_1^m(\alpha_i, \boldsymbol{\delta}_{\langle ij \rangle})$ and $V_2^{lmn}(i,j) \equiv V_2^{lmn}(\alpha_i, \boldsymbol{\delta}_{\langle ij \rangle})$ are functions of sublattice and bond. Here, $\alpha_i$ is the sublattice index of site $i$, while $\boldsymbol{\delta}_{\langle ij \rangle}$ the vector that connects sites $i$ and $j$. After performing the Fourier transform

$$
\beta_{(\alpha, \boldsymbol{q})\sigma} = N_{uc}^{-1/2} \sum_{\boldsymbol{r}} e^{-i\boldsymbol{q}\cdot\boldsymbol{r}} \beta_{(\alpha,\boldsymbol{r})\sigma},
$$

(9)

where $(\alpha, \boldsymbol{r})$ denotes the lattice site with coordinate $\boldsymbol{r}$ that belongs to sublattice $\alpha$ and $N_{uc}$ the total number of the magnetic unit cells, we obtain

$$
\mathcal{H}^{(3)} = \frac{1}{\sqrt{N_{uc}}} \sum_{\substack{\alpha_a, \boldsymbol{q}_a \in BZ \\ \sigma_a \neq 1}} \delta\left( \sum_a \boldsymbol{q}_a - \boldsymbol{G} \right) V_{\alpha_1,\alpha_2,\alpha_3}^{\sigma_1\sigma_2\sigma_3}(\boldsymbol{q}_1,\boldsymbol{q}_2,\boldsymbol{q}_3) \beta_{(\alpha_1,\bar{\boldsymbol{q}}_1),\sigma_1}^\dagger \beta_{(\alpha_2,\boldsymbol{q}_2),\sigma_2} \beta_{(\alpha_3,\boldsymbol{q}_3),\sigma_3} + \text{H.c.},
$$

(10)

with

$$
V_{\alpha_1,\alpha_2,\alpha_3}^{\sigma_1\sigma_2\sigma_3}(\boldsymbol{q}_1,\boldsymbol{q}_2,\boldsymbol{q}_3) = \sum_{\langle ij \rangle'} \tilde{V}_{\langle ij \rangle'}(1,2,3) + \sum_\alpha \tilde{V}_\alpha(1,2,3),
$$

(11)

where $1 \leq a \leq 3$, $\sum_{\langle ij \rangle'}$ sums over translationally inequivalent bonds. The first term of Eq. (11) includes the off-site or bond contributions to the cubic vertex that arise from the exchange interactions. The corresponding vertex function is

$$
\begin{aligned}
\tilde{V}_{\langle ij \rangle'}(1,2,3) = \sqrt{M} \Big[ &V_1^{\sigma_3}(\alpha_i, \boldsymbol{\delta}_{\langle ij \rangle}) \delta_{\alpha_1\alpha_j} \delta_{\alpha_2\alpha_j} \delta_{\alpha_3\alpha_j} \delta_{\sigma_1\sigma_2} \\
&+ V_1^{\sigma_3}(\alpha_j, \bar{\boldsymbol{\delta}}_{\langle ij \rangle}) \delta_{\alpha_1\alpha_i} \delta_{\alpha_2\alpha_i} \delta_{\alpha_3\alpha_i} \delta_{\sigma_1\sigma_2} \\
&+ V_2^{\sigma_3\sigma_1\sigma_2}(\alpha_i, \boldsymbol{\delta}_{\langle ij \rangle}) e^{-i\boldsymbol{q}_3 \cdot \boldsymbol{\delta}_{ij}} \delta_{\alpha_1\alpha_j} \delta_{\alpha_2\alpha_j} \delta_{\alpha_3\alpha_i} \\
&+ V_2^{\sigma_3\sigma_1\sigma_2}(\alpha_j, \bar{\boldsymbol{\delta}}_{\langle ij \rangle}) e^{i\boldsymbol{q}_3 \cdot \boldsymbol{\delta}_{ij}} \delta_{\alpha_1\alpha_i} \delta_{\alpha_2\alpha_i} \delta_{\alpha_3\alpha_j} \Big]
\end{aligned}
$$

(12)

where $\boldsymbol{\delta}_{ij} = \boldsymbol{r}_j - \boldsymbol{r}_i$ and $\bar{\boldsymbol{\delta}}_{ij} \equiv -\boldsymbol{\delta}_{ij}$ refer to the bond vectors. The second term of Eq. (11) includes the on-site contributions to the cubic vertex that arise from the single-ion anisotropy and the Zeeman term. The corresponding vertex function is

$$
\tilde{V}_\alpha(1,2,3) = \frac{1}{2\sqrt{M}} \left( D\mathcal{Q}_{1\sigma_3}^\mu(\alpha_i) + h\mathcal{S}_{1\sigma_3}^\mu(\alpha_i) \right) \delta_{\alpha_1\alpha} \delta_{\alpha_2\alpha} \delta_{\alpha_3\alpha} \delta_{\sigma_1\sigma_2},
$$

(13)

The quasiparticle modes are obtained by performing a Bogoliubov transformation

$$
\begin{pmatrix} \beta_{(\alpha,\boldsymbol{q}),\sigma} \\ \beta_{(\alpha,\bar{\boldsymbol{q}}),\sigma}^\dagger \end{pmatrix} = \begin{pmatrix} W_{(\alpha,\sigma),n}^{11}(\boldsymbol{q}) & W_{(\alpha,\sigma),n}^{12}(\boldsymbol{q}) \\ W_{(\alpha,\sigma),n}^{21}(\boldsymbol{q}) & W_{(\alpha,\sigma),n}^{22}(\boldsymbol{q}) \end{pmatrix} \begin{pmatrix} \gamma_{n,\boldsymbol{q}} \\ \gamma_{n,\bar{\boldsymbol{q}}}^\dagger \end{pmatrix},
$$

(14)

that diagonalizes the linear spin-wave Hamiltonian $\mathcal{H}^{(2)}(\boldsymbol{q})$. Note that this transformation is redundant for $\pm\boldsymbol{q}$, implying that

$$
W_{(\alpha,\sigma),n}^{11}(-\boldsymbol{q}) = W_{(\alpha,\sigma),n}^{22*}(\boldsymbol{q}), \quad W_{(\alpha,\sigma),n}^{12}(-\boldsymbol{q}) = W_{(\alpha,\sigma),n}^{21*}(\boldsymbol{q}),
$$

(15)

$$
W_{(\alpha,\sigma),n}^{21}(-\boldsymbol{q}) = W_{(\alpha,\sigma),n}^{12*}(\boldsymbol{q}), \quad W_{(\alpha,\sigma),n}^{22}(-\boldsymbol{q}) = W_{(\alpha,\sigma),n}^{11*}(\boldsymbol{q}).
$$

(16)

The triple product of bosonic operators in Eq. (10) becomes

$$
\begin{aligned}
&\beta_{(\alpha_1,\bar{\boldsymbol{q}}_1),\sigma_1}^\dagger \beta_{(\alpha_2,\boldsymbol{q}_2),\sigma_2} \beta_{(\alpha_3,\boldsymbol{q}_3),\sigma_3} \\
&= \sum_{n_1,n_2,n_3} W_{(\alpha_1,\sigma_1),n_1}^{21}(\boldsymbol{q}_1) W_{(\alpha_2,\sigma_2),n_2}^{11}(\boldsymbol{q}_2) W_{(\alpha_3,\sigma_3),n_3}^{11}(\boldsymbol{q}_3) \gamma_{n_1,\boldsymbol{q}_1} \gamma_{n_2,\boldsymbol{q}_2} \gamma_{n_3,\boldsymbol{q}_3} \\
&+ W_{(\alpha_1,\sigma_1),n_1}^{21}(\boldsymbol{q}_1) W_{(\alpha_2,\sigma_2),n_2}^{12}(\boldsymbol{q}_2) W_{(\alpha_3,\sigma_3),n_3}^{11}(\boldsymbol{q}_3) \gamma_{n_1,\boldsymbol{q}_1} \gamma_{n_2,\bar{\boldsymbol{q}}_2}^\dagger \gamma_{n_3,\boldsymbol{q}_3} \\
&+ W_{(\alpha_1,\sigma_1),n_1}^{21}(\boldsymbol{q}_1) W_{(\alpha_2,\sigma_2),n_2}^{11}(\boldsymbol{q}_2) W_{(\alpha_3,\sigma_3),n_3}^{12}(\boldsymbol{q}_3) \gamma_{n_1,\boldsymbol{q}_1} \gamma_{n_2,\boldsymbol{q}_2} \gamma_{n_3,\bar{\boldsymbol{q}}_3}^\dagger \\
&+ W_{(\alpha_1,\sigma_1),n_1}^{21}(\boldsymbol{q}_1) W_{(\alpha_2,\sigma_2),n_2}^{12}(\boldsymbol{q}_2) W_{(\alpha_3,\sigma_3),n_3}^{12}(\boldsymbol{q}_3) \gamma_{n_1,\boldsymbol{q}_1} \gamma_{n_2,\bar{\boldsymbol{q}}_2}^\dagger \gamma_{n_3,\bar{\boldsymbol{q}}_3}^\dagger \\
&+ W_{(\alpha_1,\sigma_1),n_1}^{22}(\boldsymbol{q}_1) W_{(\alpha_2,\sigma_2),n_2}^{11}(\boldsymbol{q}_2) W_{(\alpha_3,\sigma_3),n_3}^{11}(\boldsymbol{q}_3) \gamma_{n_1,\bar{\boldsymbol{q}}_1}^\dagger \gamma_{n_2,\boldsymbol{q}_2} \gamma_{n_3,\boldsymbol{q}_3} \\
&+ W_{(\alpha_1,\sigma_1),n_1}^{22}(\boldsymbol{q}_1) W_{(\alpha_2,\sigma_2),n_2}^{12}(\boldsymbol{q}_2) W_{(\alpha_3,\sigma_3),n_3}^{11}(\boldsymbol{q}_3) \gamma_{n_1,\bar{\boldsymbol{q}}_1}^\dagger \gamma_{n_2,\bar{\boldsymbol{q}}_2}^\dagger \gamma_{n_3,\boldsymbol{q}_3} \\
&+ W_{(\alpha_1,\sigma_1),n_1}^{22}(\boldsymbol{q}_1) W_{(\alpha_2,\sigma_2),n_2}^{11}(\boldsymbol{q}_2) W_{(\alpha_3,\sigma_3),n_3}^{12}(\boldsymbol{q}_3) \gamma_{n_1,\bar{\boldsymbol{q}}_1}^\dagger \gamma_{n_2,\boldsymbol{q}_2} \gamma_{n_3,\bar{\boldsymbol{q}}_3}^\dagger \\
&+ W_{(\alpha_1,\sigma_1),n_1}^{22}(\boldsymbol{q}_1) W_{(\alpha_2,\sigma_2),n_2}^{12}(\boldsymbol{q}_2) W_{(\alpha_3,\sigma_3),n_3}^{12}(\boldsymbol{q}_3) \gamma_{n_1,\bar{\boldsymbol{q}}_1}^\dagger \gamma_{n_2,\bar{\boldsymbol{q}}_2}^\dagger \gamma_{n_3,\bar{\boldsymbol{q}}_3}^\dagger.
\end{aligned}
$$

(17)

After putting the $\gamma$ operators in normal ordering and ignoring the linear terms that arise from this process, which is justified because of the strong easy-axis anisotropy, we obtain

$$
\begin{aligned}
&V_{\alpha_1,\alpha_2,\alpha_3}^{\sigma_1\sigma_2\sigma_3}(\boldsymbol{q}_1,\boldsymbol{q}_2,\boldsymbol{q}_3) \beta_{(\alpha_1,\bar{\boldsymbol{q}}_1),\sigma_1}^\dagger \beta_{(\alpha_2,\boldsymbol{q}_2),\sigma_2} \beta_{(\alpha_3,\boldsymbol{q}_3),\sigma_3} + \text{H.c.} \\
&= \sum_{\{n_i\}} \tilde{V}_{n_1n_2n_3}^{(1)}(\boldsymbol{q}_1,\boldsymbol{q}_2,\boldsymbol{q}_3) \gamma_{n_1,\bar{\boldsymbol{q}}_1}^\dagger \gamma_{n_2,\boldsymbol{q}_2} \gamma_{n_3,\boldsymbol{q}_3} + \sum_{\{n_i\}} \tilde{V}_{n_1n_2n_3}^{(2)}(\boldsymbol{q}_1,\boldsymbol{q}_2,\boldsymbol{q}_3) \gamma_{n_1,\bar{\boldsymbol{q}}_1}^\dagger \gamma_{n_2,\bar{\boldsymbol{q}}_2}^\dagger \gamma_{n_3,\bar{\boldsymbol{q}}_3}^\dagger + \text{H.c.},
\end{aligned}
$$

(18)

where

$$
\begin{aligned}
&\tilde{V}_{n_1n_2n_3}^{(1)}(\boldsymbol{q}_1,\boldsymbol{q}_2,\boldsymbol{q}_3) \\
&= \sum_{\{\alpha_i,\sigma_i\}} V_{\alpha_1,\alpha_2,\alpha_3}^{\sigma_1\sigma_2\sigma_3}(\boldsymbol{q}_1,\boldsymbol{q}_2,\boldsymbol{q}_3) W_{(\alpha_1,\sigma_1),n_1}^{22}(\boldsymbol{q}_1) W_{(\alpha_2,\sigma_2),n_2}^{11}(\boldsymbol{q}_2) W_{(\alpha_3,\sigma_3),n_3}^{11}(\boldsymbol{q}_3) \\
&+ V_{\alpha_1,\alpha_2,\alpha_3}^{\sigma_1\sigma_2\sigma_3}(\boldsymbol{q}_3,\boldsymbol{q}_2,\boldsymbol{q}_1) W_{(\alpha_1,\sigma_1),n_3}^{21}(\boldsymbol{q}_3) W_{(\alpha_2,\sigma_2),n_2}^{11}(\boldsymbol{q}_2) W_{(\alpha_3,\sigma_3),n_1}^{12}(\boldsymbol{q}_1) \\
&+ V_{\alpha_1,\alpha_2,\alpha_3}^{\sigma_1\sigma_2\sigma_3}(\boldsymbol{q}_2,\boldsymbol{q}_1,\boldsymbol{q}_3) W_{(\alpha_1,\sigma_1),n_2}^{21}(\boldsymbol{q}_2) W_{(\alpha_2,\sigma_2),n_1}^{12}(\boldsymbol{q}_1) W_{(\alpha_3,\sigma_3),n_3}^{11}(\boldsymbol{q}_3) \\
&+ V_{\alpha_1,\alpha_2,\alpha_3}^{\sigma_1\sigma_2\sigma_3*}(\bar{\boldsymbol{q}}_1,\bar{\boldsymbol{q}}_2,\bar{\boldsymbol{q}}_3) W_{(\alpha_1,\sigma_1),n_1}^{21*}(\bar{\boldsymbol{q}}_1) W_{(\alpha_2,\sigma_2),n_2}^{12*}(\bar{\boldsymbol{q}}_2) W_{(\alpha_3,\sigma_3),n_3}^{12*}(\bar{\boldsymbol{q}}_3) \\
&+ V_{\alpha_1,\alpha_2,\alpha_3}^{\sigma_1\sigma_2\sigma_3*}(\bar{\boldsymbol{q}}_3,\bar{\boldsymbol{q}}_2,\bar{\boldsymbol{q}}_1) W_{(\alpha_1,\sigma_1),n_3}^{22*}(\bar{\boldsymbol{q}}_3) W_{(\alpha_2,\sigma_2),n_2}^{12*}(\bar{\boldsymbol{q}}_2) W_{(\alpha_3,\sigma_3),n_1}^{11*}(\bar{\boldsymbol{q}}_1) \\
&+ V_{\alpha_1,\alpha_2,\alpha_3}^{\sigma_1\sigma_2\sigma_3*}(\bar{\boldsymbol{q}}_3,\bar{\boldsymbol{q}}_1,\bar{\boldsymbol{q}}_2) W_{(\alpha_1,\sigma_1),n_3}^{22*}(\bar{\boldsymbol{q}}_3) W_{(\alpha_2,\sigma_2),n_1}^{11*}(\bar{\boldsymbol{q}}_1) W_{(\alpha_3,\sigma_3),n_2}^{12*}(\bar{\boldsymbol{q}}_2),
\end{aligned}
$$

and

$$
\begin{aligned}
\tilde{V}_{n_1n_2n_3}^{(2)}(\boldsymbol{q}_1,\boldsymbol{q}_2,\boldsymbol{q}_3) = &\sum_{\{\alpha_i,\sigma_i\}} V_{\alpha_1,\alpha_2,\alpha_3}^{\sigma_1\sigma_2\sigma_3}(\boldsymbol{q}_1,\boldsymbol{q}_2,\boldsymbol{q}_3) W_{(\alpha_1,\sigma_1),n_1}^{22}(\boldsymbol{q}_1) W_{(\alpha_2,\sigma_2),n_2}^{12}(\boldsymbol{q}_2) W_{(\alpha_3,\sigma_3),n_3}^{12}(\boldsymbol{q}_3) \\
&+ V_{\alpha_1,\alpha_2,\alpha_3}^{\sigma_1\sigma_2\sigma_3*}(\bar{\boldsymbol{q}}_3,\bar{\boldsymbol{q}}_2,\bar{\boldsymbol{q}}_1) W_{(\alpha_1,\sigma_1),n_3}^{21*}(\bar{\boldsymbol{q}}_3) W_{(\alpha_2,\sigma_2),n_2}^{11*}(\bar{\boldsymbol{q}}_2) W_{(\alpha_3,\sigma_3),n_1}^{11*}(\bar{\boldsymbol{q}}_1).
\end{aligned}
$$

(19)

The final form of the cubic interaction is obtained after symmetrization of the vertex:

$$
\begin{aligned}
\mathcal{H}^{(3)} = \frac{1}{\sqrt{N_{uc}}} \sum_{\substack{\alpha_a, \boldsymbol{q}_a \in BZ \\ \sigma_a \neq 1}} \delta\left( \sum_a \boldsymbol{q}_a - \boldsymbol{G} \right) &\Bigg[ \frac{1}{2!} V_{n_1n_2n_3}^{(S1)}(\boldsymbol{q}_1,\boldsymbol{q}_2,\boldsymbol{q}_3) \gamma_{n_1,\bar{\boldsymbol{q}}_1}^\dagger \gamma_{n_2,\boldsymbol{q}_2} \gamma_{n_3,\boldsymbol{q}_3} \\
&+ \frac{1}{3!} \sum_{\{n_i,\boldsymbol{q}_i\}} \tilde{V}_{n_1n_2n_3}^{(S2)}(\boldsymbol{q}_1,\boldsymbol{q}_2,\boldsymbol{q}_3) \gamma_{n_1,\bar{\boldsymbol{q}}_1}^\dagger \gamma_{n_2,\bar{\boldsymbol{q}}_2}^\dagger \gamma_{n_3,\bar{\boldsymbol{q}}_3}^\dagger \Bigg] + \text{H.c.},
\end{aligned}
$$

(20)

where

$$\tilde{V}^{(S1)}_{n_1 n_2 n_3}(\boldsymbol{q}_1,\boldsymbol{q}_2,\boldsymbol{q}_3) = \sum_{P(2,3)} \tilde{V}^{(1)}_{n_1 n_2 n_3}(\boldsymbol{q}_1,\boldsymbol{q}_2,\boldsymbol{q}_3),$$

$$\tilde{V}^{(S2)}_{n_1 n_2 n_3}(\boldsymbol{q}_1,\boldsymbol{q}_2,\boldsymbol{q}_3) = \sum_{P(1,2,3)} \tilde{V}^{(2)}_{n_1 n_2 n_3}(\boldsymbol{q}_1,\boldsymbol{q}_2,\boldsymbol{q}_3),$$

(21)

and $P$ the permutation operator.

To compare with the inelastic neutron-scattering data, we compute the dynamical spin structure factor at zero temperature, $S_{\mu\nu}(\mathbf{q},\omega) = 2\Theta(\omega)\chi''_{\mu\nu}(\mathbf{q},\omega)$, where $\Theta(\omega)$ is the Heaviside step function and $\chi''_{\mu\nu}(\mathbf{q},\omega)$ is the imaginary part of the dynamical spin susceptibility

$$i\chi_{\mu\nu}(\mathbf{q},\omega) = \frac{1}{4}\sum_{\alpha\beta}\int_0^\infty dt e^{i\omega t}\left\langle\left[S^\mu_{\alpha,\boldsymbol{q}}(t),S^\nu_{\beta,-\boldsymbol{q}}(0)\right]\right\rangle,$$

(22)

where $S^\mu_{\alpha,\boldsymbol{q}} = N_{uc}^{-1/2}\sum_{\boldsymbol{r}} e^{-i\boldsymbol{q}\cdot\boldsymbol{r}}S^\mu_{\alpha,\boldsymbol{r}}$. $\chi_{\mu\nu}(\boldsymbol{q},\omega)$ was evaluated at zero magnetic field before[24] at the linear level, i. e., without including the effect of the interaction term $\mathcal{H}^{(3)}$ in Eq. (7). We note that the longitudinal channel of the spin structure factor has only contributions from the two-magnon continuum, which are negligibly small for FeI$_2$ because of the strong single-ion anisotropy. We will then focus on the transverse response and on the effects produced by the interactions between quasiparticles. The key observation is that the kinematic conditions for magnon decay become satisfied for certain ranges of magnetic field values, giving rise to an intrinsic broadening or finite lifetime of the corresponding quasiparticle.

To leading order in $1/M$, the dynamical spin susceptibility is given by

$$\chi_{\alpha\beta}^{\mu\nu}(\mathbf{q},\omega) = M\sum_{mn}\begin{pmatrix} S^\mu_{1m}(\alpha) \\ S^\mu_{m1}(\alpha) \end{pmatrix}^T \begin{pmatrix} \mathcal{G}'(\boldsymbol{q},\omega) & \breve{\mathcal{G}}(\boldsymbol{q},\omega) \\ \hat{\mathcal{G}}(\boldsymbol{q},\omega) & \mathcal{G}''(\boldsymbol{q},\omega) \end{pmatrix}_{(\alpha,m)(\beta,n)}\begin{pmatrix} S^\nu_{n1}(\beta) \\ S^\nu_{1n}(\beta) \end{pmatrix},$$

(23)

where the $2\times2$ block matrix $\mathcal{G}(\boldsymbol{q},\omega)$ is the *interacting* single-particle Green's function determined by the Dyson equation

$$\mathcal{G}^{-1}(\boldsymbol{q},\omega) = \mathcal{G}_0^{-1}(\boldsymbol{q},\omega) - \Sigma(\boldsymbol{q},\omega).$$

(24)

The *non-interacting* single-particle Green's function $\mathcal{G}_0(\boldsymbol{q},\omega)$ is given by

$$\begin{pmatrix} \mathcal{G}'_0(\boldsymbol{q},\omega) & \breve{\mathcal{G}}_0(\boldsymbol{q},\omega) \\ \hat{\mathcal{G}}_0(\boldsymbol{q},\omega) & \mathcal{G}''_0(\boldsymbol{q},\omega) \end{pmatrix} = \left(-(\omega+i0^+)A + \mathcal{H}^{(2)}\right)^{-1},$$

(25)

where

$$A = \begin{pmatrix} I_{8\times8} & 0 \\ 0 & -I_{8\times8} \end{pmatrix}$$

(26)

and $I_{8\times8}$ is the $8\times8$ identity matrix. Given that $\mathcal{H}^{(2)}$ is of order $M$ and the energy scale of interest is $\omega \propto (M)^1$, we obtain that $\mathcal{G}_0$ is of order $M^{-1}$, as it was mentioned in the previous section. The single-particle self-energy, $\Sigma(\boldsymbol{q},\omega)$, is given by the two one-loop diagrams:

that correspond to the self-energy corrections:

$$\Sigma^{(a)}_{n_1}(\boldsymbol{q},\omega) = \frac{1}{2N_{uc}}\sum_{\boldsymbol{k},n_2 n_3}\frac{|\tilde{V}^{(S1)}_{n_1 n_2 n_3}(\bar{\boldsymbol{q}},\boldsymbol{k},\boldsymbol{q}-\boldsymbol{k})|^2}{\omega - \varepsilon_{n_2,\boldsymbol{k}} - \varepsilon_{n_3,\boldsymbol{q}-\boldsymbol{k}} + i0^+},$$

(27)

and

$$\Sigma^{(b)}_{n_1}(\boldsymbol{q},\omega) = -\frac{1}{2N_{uc}}\sum_{\boldsymbol{k},n_2 n_3}\frac{|\tilde{V}^{(S2)}_{n_1 n_2 n_3}(\bar{\boldsymbol{q}},\boldsymbol{k},\boldsymbol{q}-\boldsymbol{k})|^2}{\omega + \varepsilon_{n_2,\boldsymbol{k}} + \varepsilon_{n_3,\boldsymbol{q}-\boldsymbol{k}} - i0^+},$$

(28)

where $\varepsilon_{n,\boldsymbol{k}}$ is the linear spin-wave dispersion. Since we are working to the leading order in $1/M$, it is enough to just consider diagonal elements where the initial and the final boson belong to the same single-particle band. For each band, the self-energy is evaluated in the on-shell approximation: $\omega = \varepsilon_{n_1,\boldsymbol{q}}$.

## Diagonalization

To study the non-perturbative effect, e.g., the avoided-decay of excitations observed at $\mu_0 H = 3$T, we performed an exact diagonalization (ED) study in the truncated subspace $\mathcal{S}_{1,2}$ with number of quasiparticles $\leq2$ on a finite lattice of $5\times5\times5$ unit cells (500 spins). As the Hilbert space dimension becomes prohibitively large for ED if we include states with three quasiparticles, our calculation only accounts for 1-, 2-, and 4-magnon excitations. The excluded states have only perturbative effects because of their higher-energy scales compared to that of a single quasiparticle.

The subspace $\mathcal{S}_{1,2}$ is spanned by the basis $\{|i\rangle,|i\leq j\rangle\}$ with $|i\rangle = \gamma_i^\dagger|\emptyset\rangle$ and $|i\leq j\rangle = \zeta_{ij}\gamma_i^\dagger\gamma_j^\dagger|\emptyset\rangle$, where $i$ stands for the dictionary index of $(n_i,\boldsymbol{q}_i)$, $|\emptyset\rangle$ refers to the vacuum of the $\gamma$-quasiparticles and $\zeta_{i\neq j} = 1$ and $\zeta_{i=j} = 1/\sqrt{2!}$ are normalization factors. Introducing the projector $\mathcal{P}_{1,2}$ to the subspace $\mathcal{S}_{1,2}$, the restricted Hamiltonian is obtained by the projection

$$\mathcal{P}_{1,2}\mathcal{H}\mathcal{P}_{1,2} = \begin{pmatrix} \mathcal{H}_{11} & \mathcal{H}_{12} \\ h.c. & \mathcal{H}_{22} \end{pmatrix},$$

(29)

with matrix elements

$$\mathcal{H}_{11}^{i,j} = \delta_{ij}\varepsilon_{n_i,\boldsymbol{q}_i}, \quad \mathcal{H}_{12}^{i,j\leq k} = \frac{1}{\sqrt{N_{uc}}}V^{(S1)}_{n_i,n_j,n_k}(\boldsymbol{q}_i,\boldsymbol{q}_j,\boldsymbol{q}_k)\zeta_{j,k},$$

$$\mathcal{H}_{22}^{i\leq j,k\leq l} = \delta_{ik}\delta_{jl}(\varepsilon_{n_i,\boldsymbol{q}_i} + \varepsilon_{n_j,\boldsymbol{q}_j})\zeta_{ij}^2 + \frac{1}{N_{uc}}U_{n_i,n_j,n_k,n_l}(\boldsymbol{q}_i,\boldsymbol{q}_j,\boldsymbol{q}_k,\boldsymbol{q}_l)\zeta_{ij}\zeta_{kl}.$$

(30)

Here, the function $U_{n_i,n_j,n_k,n_l}(\boldsymbol{q}_i,\boldsymbol{q}_j,\boldsymbol{q}_k,\boldsymbol{q}_l)$ accounts for the interaction between $\gamma$-quasiparticles,

$$\mathcal{H}^{(4)} = \frac{1}{2!2!N_{uc}}\sum_{ijkl}\delta(\boldsymbol{q}_i+\boldsymbol{q}_j+\boldsymbol{q}_k+\boldsymbol{q}_l-\boldsymbol{G})U_{n_i,n_j,n_k,n_l}(\boldsymbol{q}_i,\boldsymbol{q}_j,\boldsymbol{q}_k,\boldsymbol{q}_l)\gamma^\dagger_{n_i,\bar{\boldsymbol{q}}_i}\gamma^\dagger_{n_j,\bar{\boldsymbol{q}}_j}\gamma_{n_k,\boldsymbol{q}_k}\gamma_{n_l,\boldsymbol{q}_l},$$

(31)

where $\boldsymbol{G}$ is the reciprocal lattice vector. The diagonalization of $\mathcal{P}_{1,2}\mathcal{H}\mathcal{P}_{1,2}$ is done for a fixed center of mass momentum. It reveals that the spectrum includes four energy levels below the two-particle continuum with a strong 4-magnon character, which are identified as the 4-magnon bound states.

The spectral weights carried by these excitations are revealed by computing the dynamical spin structure factor within the subspace

$\mathcal{S}_{1,2}$:

$$S_{\mu\nu}(\boldsymbol{q},\omega) = \int_{-\infty}^{\infty} dt e^{i\omega t} \frac{1}{N} \sum_{ij} e^{-i\boldsymbol{q}\cdot(\boldsymbol{r}_j - \boldsymbol{r}_i)} \left\langle \emptyset | S^{\mu}_{\boldsymbol{r}_j}(t) S^{\nu}_{\boldsymbol{r}_i}(0) | \emptyset \right\rangle$$

$$= \int_{-\infty}^{\infty} dt e^{i\omega t} \frac{1}{4} \sum_{\alpha\beta} \left\langle \emptyset | S^{\mu}_{\alpha,\boldsymbol{q}}(t) S^{\nu}_{\beta,-\boldsymbol{q}}(0) | \emptyset \right\rangle, \qquad (32)$$

where $N = 4N_{uc}$ is the total number of lattice sites, and

$$S^{\mu}_{\alpha,\boldsymbol{q}}(t) = (1/\sqrt{N_{uc}}) \sum_{\boldsymbol{r} \in \alpha} e^{-i\boldsymbol{q}\cdot\boldsymbol{r}} S^{\mu}_{\boldsymbol{r}}(t) \qquad (33)$$

is the Fourier transform of the spin operators on the sublattice $\alpha$. The evaluation of this correlation function is carried out by using the continued-fraction method[43] based on the Lanczos algorithm[44]. The lattice size employed here ($5 \times 5 \times 5$ unit cells) is large enough to capture the 4-magnon bound states because their linear size is of the order of one lattice space owing to the very large effective mass of the two-magnon bound sates.

## Data availability
The raw experimental data are stored on ORNL's Neutron-Scattering Division computers. The reduced experimental data and the theory dataaa are available from the corresponding authors under simple requests.

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

## Acknowledgements

We thank Tyrel McQueen for his help with crystal growth at PARADIM. The work of X.B., Z.L.D., and M.M. at Georgia Tech was supported by the U.S. Department of Energy, Office of Science, Basic Energy Sciences, Materials Sciences and Engineering Division under award DE-SC-0018660. The work of H.Z. at the Oak Ridge National Laboratory was supported by the U.S. Department of Energy, Office of Science, Basic Energy Sciences, Materials Sciences and Engineering Division. The work of S.-S.Z. and C.D.B. at the University of Tennessee was supported by the Lincoln Chair of Excellence in Physics. Growth of $FeI_2$ crystals was supported by the National Science Foundation's PARADIM (Platform for the Accelerated Realization, Analysis, and Discovery of Interface Materials) under Cooperative Agreement No. DMR-1539918. Some of this work were performed in part at the Materials Characterization Facility at Georgia Tech that is jointly supported by the GT Institute for Materials and the Institute for Electronics and Nanotechnology, which is a member of the National Nanotechnology Coordinated Infrastructure supported by the National Science Foundation under Grant No. ECCS-2025462. This research used resources at the High Flux Isotope Reactor and Spallation Neutron Source, a DOE Office of Science User Facility operated by the Oak Ridge National Laboratory.

## Author contributions

X.B., M.M., and C.D.B. conceived the project. Z.L.D., X.B., and W.A.P grew the sample at the PARADIM facility. X.B., V.O.G., and M.M. performed the neutron-scattering measurements. X.B. analyzed the neutron-scattering data. S.-S.Z., H.Z., and C.D.B. carried out the theoretical calculations. X.B., S.-S.Z., M.M., and C.D.B. wrote the manuscript with input from all authors.

## Competing interests

The authors declare no competing interests.
