## [Peer Review File · Nature Communications]

REVIEWERS' COMMENTS

Reviewer #1 (Remarks to the Author):

In this work, experimental and theoretical results for the excitation spectra of a spin system (Fe₂) are reported.

Even though the outcome of this work can be interesting for people working on specific aspects of frustrated magnetism, I do not think that it is not sufficiently original and relevant to be published in *Nature Communications*. Indeed, the paper is largely oversold in the introduction and conclusions, with references to high-energy particle physics. Instead, they just present a combined study that shows the presence of both magnons and what they call "single-ion bound states". The fact that they interact with each other and determine decay processes is not surprising, though never discussed in detail for a specific material (magnon instabilities have been already observed in several papers by now in several materials [Nat. Phys. 11, 62 (2015)]). Still, I do not think that these facts justify a publication in a high-impact journal.

From a specific aspect, I would expect a large 2-magnon continuum for any value of the external magnetic field and I would like to understand how it is possible to have such a well defined excitation, which should be well inside the continuum. Related to that, what is the reason for having only limited regions of finite decay rate in E₃,...,E₈ modes (Fig.3), while being clearly inside the continuum.

In summary, I think that the paper is not suitable to *Nature Communication*.

Reviewer #2 (Remarks to the Author):

This manuscript is a very valuable contribution to our understanding of magnetic excitations in solid matter. The authors significantly improved the manuscript from the originally submitted version, and made it much more accessible.

I recommend immediate publication in its present form.

Re-Submission of “Instabilities of heavy magnons in an anisotropic magnet” to Nature Communications.

We have worked diligently and modified our manuscript accordingly. Furthermore, we address the final comments of Referee #1, below:

***Ref 1:** From a specific aspect, I would expect a large 2-magnon continuum for any value of the external magnetic field, and I would like to understand how it is possible to have such a well-defined excitations, which should be well inside the continuum. Related to that, what is the reason for having only limited regions of finite decay rate in E_3, \dots, E_8 modes (Fig.3), while being clearly inside the continuum.*

Our response: We thank the referee for their continuous engagement with our manuscript. The reason it is possible to have well-defined multi-magnon excitations is because bound states are stabilized *below* the continuum by short-range ferromagnetic interactions. In zero field, all the multi-particle continua are essentially above the single-magnon (SM) and single-ion-bound-state (SIBS) excitations we are studying here, except for branch 8 which is marginally touching the multi-particle continuum (See Fig. 3d). We have added a sentence to this effect in the manuscript (in blue). As the referee points out, this situation changes in applied magnetic field with SM and SIBS modes clearly in the multi-particle continua. However, the regions with large-enough decay rates remain limited because fulfilling kinematic conditions is a necessary but not sufficient condition for spontaneous decay: the size of the matrix element also matters. Given the finite energy resolution of our experiment, decays are only visible when strongest in a narrow field range around 4T. This discussion is made crisp by inspection of Fig. 3d. We have strengthened a sentence in the manuscript to make it clear (in red).